# Effect of Source Mispositioning on the Repeatability of 4D Vertical Seismic Profiling Acquired with Distributed Acoustic Sensors

**DOI:** 10.3390/s22249742

**Published:** 2022-12-12

**Authors:** Roman Isaenkov, Konstantin Tertyshnikov, Alexey Yurikov, Pavel Shashkin, Roman Pevzner

**Affiliations:** Centre for Exploration Geophysics, Curtin University, GPO Box U1987, Perth, WA 6845, Australia

**Keywords:** DAS, time-lapse seismic, VSP, repeatability, mispositioning

## Abstract

Vertical seismic profiling (VSP) with distributed acoustic sensing (DAS) is an increasingly popular evolving technique for reservoir monitoring. DAS technology enables permanent fibre installations in wells and simultaneous seismic data recording along an entire borehole. Deploying the receivers closer to the reservoir allows for better detectability of smaller signals. A high level of repeatability is essential for the robust time-lapse monitoring of geological reservoirs. One of the prominent factors of repeatability degradation is a shift between source/receiver locations (mispositioning) during baseline and monitor surveys. While the mispositioning effect has been extensively studied for surface 4D seismic, the number of such studies for VSP is quite limited. To study the effects of source mispositioning on time-lapse data repeatability, we performed two VSP experiments at two on-shore sites with vibroseis. The first study was carried out at the Otway International Test Centre during Stage 3 of the Otway project and showed that the effect of source mispositioning on repeatability is negligible in comparison with the effect of temporal variations of the near-surface conditions. To avoid these limitations, we conducted a same-day controlled experiment at the Curtin University site. This second experiment showed that the effect of source mispositioning on repeatability is controlled by the degree of lateral variations of the near-surface conditions. Unlike in marine seismic measurements, lateral variations of near-surface properties can be strong and rapid and can degrade the repeatability for shifts of the source of a few meters. The greater the mispositioning, the higher the chance of such significant variations. When the near-surface conditions are laterally homogeneous, the effect of typical source mispositioning is small, and in all practical monitoring applications its contribution to non-repeatability is negligible.

## 1. Introduction

Time-lapse seismic measurement is an important tool for monitoring underground processes such as oil/gas reservoir production or CO_2_ geosequestration. This importance stems from the fact that the time-lapse seismic method has superior spatial resolution over other remote sensing methods [1,2]. Distributed acoustic sensing (DAS) technology has disruptive potential in seismic monitoring enabling the use of a fibre optic cable as an array of multiple seismic receivers [3]. A borehole equipped with fibre optic cable represents a dense and sensitive seismic receiver array. The introduction of optical sensors for borehole seismic measurement makes this technology the most effective for underground monitoring of the near-well medium.

Time-lapse DAS VSP has been applied for various geological tasks in industrial and research carbon dioxide geosequestration projects [4,5,6,7], water sweep monitoring in oil reservoirs, and in experiments on the shallow release of CO_2_ [8]. Nevertheless, some projects have encountered several challenges, mainly related to non-optimal acquisition parameters, a weak time-lapse signal from the reservoir, and/or a high level of time-lapse noise. The level of time-lapse noise is an essential consideration for successful monitoring projects, and is affected by different factors. Some factors are determined by nature (e.g., near-surface conditions), whereas others are influenced by acquisition design (e.g., equipment’s noise level, accurate positioning of sensors and sources etc.). Some of the factors are listed below [1]:Acquisition geometry differences
Source, receiver mispositioningSource/receiver orientation
Near-surface conditions
Near-surface variationsSource/receiver couplingEnvironment
Soil moistureGroundwater levelVegetationNoise
Ambient noiseShot-generated noiseGeology
Shallow gasSteep dipsFault shadows

In this paper, we focus on the effect of mispositioning—the spatial shift between sources/receivers’ locations between the baseline and monitor surveys. Mispositioning is otherwise known as positioning difference [9]. In the case of VSP with permanently installed sensors (e.g., cemented behind the casing, deployed on tubing), the shift between receiver locations is not a concern as they are fixed in space. Still, the accurate placement of active sources in each vintage should be ensured.

Many researchers have studied the effect of acquisition parameters on repeatability for surface land [10,11,12] and offshore seismic measurements [13,14,15]. However, the number of such publications on time-lapse VSP data is limited. Landro [16] published the most detailed study of the mispositioning effect on marine VSP time-lapse data quality, where data were acquired with a string of geophones located at 2000 m depth with a relatively low frequency (up to 50 Hz) airgun seismic source. The study showed that a mispositioning of up to 5 to 10 m has a minor consequence on repeatability. However, these results cannot be directly extrapolated to the VSP data acquired on land, primarily due to strong spatial variation in near-surface conditions.

In our study, we used two research sites to assess the effect of mispositioning on repeatability. The first one was the Otway International Test Centre (Victoria, Australia). During Stage 3 of the Otway project, three 4D DAS VSP surveys were acquired. The surveys were separated by a few months, and several shot points had to be shifted to 0.5–3 m due to changes in land access. We assessed how these shifts affected seismic repeatability and observed the strong effect of seasonal variations on repeatability, which masks the misposition effect. Thus, we designed the second experiment to avoid the effect of seasonal variation. At the second site, the Curtin/NGL research facility, we designed a same-day dedicated survey to avoid weather/seasonal impact and simulated misposition of a vibroseis source in the range of ~0.5–20 m. While the effect of mispositioning on repeatability was detectable, spatial variations of near-surface conditions dominated in the resulting repeatability change.

## 2. Case Study 1: 4D DAS VSP for CO_2_ Geosequestration Monitoring at the Otway International Test Centre

VSP experiments at the Otway International Test Centre have been an integral part of the CO2CRC Otway geosequestration project from its very start. Zero-offset VSP, offset VSP, and 4D VSP surveys were carried out in 2007–2010 during Stage 1 of the Otway Project [17] in the CRC1 and Naylor-1 wells to monitor gas injected into Waare C formation at ~2 km depth with a standard geophone tool. The first effort to detect the injected substance was ineffective, as the supercritical mixture of CO_2_-CH_4_ was injected into the depleted reservoir with some residual saturation, resulting in a very small time-lapse signal. Further borehole seismic experiments conducted during Stage 2C (2015–2018) successfully detected as little as 5 kt of CO_2_ using 4D VSP with geophones [18]. Stage 2C experiments also included a comparison of the sensitivity of DAS versus geophones [19,20] and experiments with permanent seismic sources [21]. These results were used to design the Stage 3 monitoring program, which comprises DAS VSP technology in combination with permanent surface sources [22,23] and conventional moving sources [6,24].

### 2.1. Experiment Design

The data for this study were acquired during baseline and monitor surveys for CO_2_ geosequestration monitoring during the Stage 3 Otway Project [25]. The first baseline 4D VSP survey (M6) was acquired in March 2020, and the first monitor survey (M7) occurred in January 2021. In each survey, shot points were acquired using a vibroseis source (INOVA UniVib 26,000 lbs) and five ~1600 m deep wells (CRC3-CRC7) instrumented with an engineered single-mode optical fibre cemented behind the casing (Figure 1a and Figure 2). Data were recorded using a Silixa iDAS v3 Carina unit with a 10 m gauge and 5 m pulse length. The vibroseis sweep parameters were a linear sweep, a frequency range 6-150 Hz, a sweep length of 24 s, cosine tapers of 0.5 s, and a listening time of 6 s. Data were recorded with a 1 ms temporal sampling rate and 1 m channel spacing along the well. The survey parameters are summarised in Table 1.

Due to the local farm activities during the first monitor M7, twelve shot points were shifted by ~1–2.5 m along the shot Line 13 (Figure 2). The mispositioning between most of the shots was below 0.25 m on Line 13 (Figure 2b). Shifts exceeding 0.5 m were included in this study to assess the deterioration of borehole seismic survey repeatability.

The actual shot location can differ from GPS measurement for up to 30 cm. That is because the vibroseis plate generally lands within 20–30 cm of the marked shot point location, and the differential GPS accuracy is within tens of cm. These errors increase uncertainty in measured mispositioning for up to 30 cm.

### 2.2. Effect of Mispositioning on Repeatability

To numerically evaluate the effect of mispositioning, we used normalised root mean square measure (NRMS), which for a pair of baseline and monitor traces is estimated as follows:(1)$$\mathrm{NRMS}=200\%\cdot \frac{\mathrm{RMS}\left(\mathrm{BS}-\mathrm{MT}\right)}{\mathrm{RMS}\left(\mathrm{BS}\right)+\mathrm{RMS}\left(\mathrm{MT}\right)}$$
where BS and MT are baseline and monitor traces, and RMS is a root mean square operator [26].

For the repeatability analysis, we used data recorded in the CRC4 well. The recorded data had a good signal-to-noise ratio as the well is the closest to Line 13. Thus, random noise would not mask the mispositioning effect.

To study repeatability, we applied minimal processing prior to the data. Recorded data were decimated to a 2 ms sample rate and 5 m spatial channel sampling. The data were correlated with a single theoretical pilot vibroseis sweep. The temporal decimation should not degrade the data quality as the recorded frequencies were below 250 Hz, and the utilised engineered DAS fibre was designed for 5 m spacing. An example of initial data repeatability assessment is illustrated in Figure 3.

Three baseline shot points (Figure 3a–c), their time-lapse differences (Figure 3d–f) and NRMS maps (Figure 3g–i) had mispositioning of 0.1, 0.2 and 2.4 m, denoted as A, B and C, respectively. NRMS plots were computed using a 60 ms running average window. In this example, shot point A (SP A) had the best repeatability with a misposition of 0.1 m, and NRMS level was primarily within 30–75% in a 400 ms window starting from direct wave (the region between red and blue dashed lines). SP B, with a low misposition of 0.2 m, and SP C, with a high 2.4 m mispositioning, had similar and relatively low repeatability within the 75–100% NRMS range. One can see that a pair of shots with significant misposition (>1 m) may have the same repeatability level as a pair with almost identical acquisition geometry (<0.3 m). In this case, the decrease in repeatability for SP B compared to SP A was likely to be caused by seasonal variation in near-surface conditions between baseline and monitor surveys.

To assess whether misposition has a major or minor effect on repeatability, we estimated average repeatability for every shot point on Line 13. Average repeatability was estimated within two windows: window A was a small 60 ms window around the direct P wave (±30 ms around the red dashed line Figure 3), and the larger 400 ms window B, which included primary, reflected, and multiple P wave reflections (400 ms down from the direct P wave, or the window between red and blue dashed lines Figure 3).

The average repeatability for both window A and window B strongly depended on the offset between a shot point and the well because the signal-to-noise ratio decreases with the distance. As the CRC4 well was deviated by 20° and had the lateral offset from the wellhead in the North-East direction of about 400 m, the shot points west from the well location (Figure 4, from −400 to 0 m) had a lower signal-to-noise ratio due to the distance and DAS directional sensitivity, resulting in poorer repeatability. This is because a straight DAS cable is not sensitive to P waves approaching the fibre at near-normal incidence [19] within the deviated part of the well. Conversely, shot points located above and east of the well had a higher signal-to-noise ratio and better repeatability. As such, for analysis, we chose offsets within the range of 0–400 m to minimise the effect of random noise on repeatability.

The average repeatability for the changed geometry (mispositioning >0.5 m) and repeated geometry (mispositioning <0.5 m) was very similar (Figure 4). The average NRMS for window A was 48%, while shifted shots had an increase in NRMS of only 4%. The same observation was made for a larger window B: repeated shot points had 79.5% NRMS, while shifted shot points had 81.5% NRMS.

In this example, the datasets were acquired at different times of the year. It is evident that seasonal near-surface variations affected repeatability more significantly than positioning errors. The spread of NRMS for twelve consecutive shifted shots varied from about 60% to 100% NRMS, and for repeated shot points from 20 to 120%. In contrast, the difference in 2–4% NRMS created by mispositioning was negligible and statistically insignificant.

There are several considerations in this study that have to be noted. First, the study included a limited number of shot points: 110 shots in total, with 12 shots being mispositioned. The observed difference between repeated and shifted shot points was relatively small compared to the repeatability variance. Secondly, the major repeatability variations were likely to be linked to the seasonal effect of near-surface variations, as the data were acquired during different seasons [27]. Decoupling the mispositioning and seasonal variations effects was challenging. The main conclusion from this study for the Otway site is that the impact of mispositioning is negligible if shot points shifted for less than 1–2 m compared to the effects of seasonal near-surface variations.

## 3. Case Study 2: Controlled Experiment at Curtin Research Facility

To exclude the effect of seasonal near-surface variations, we conducted a controlled experiment at the Curtin/NGL research facility (Perth, Western Australia) (Figure 1b). We designed and acquired the DAS VSP survey using a vibroseis source (INOVA UniVib 26,000 lbs) and a fibre optic cable installed behind the casing in a 900 m deep well. Source points simulating the changes in acquisition geometry formed an eight-azimuth asterisk-shaped pattern (Figure 5). The central shot point was 130 m away from the wellhead. The length of each line was 10–15 m with 8–10 shot points per line starting from the centre. The central shot point was repeated eight times with different orientations of the vibroseis truck. The total number of shot points was 76. The distance between shot points was as small as 0.5 m (half the size of the vibroseis plate) near the centre of the asterisk and increased with the distance from the centre. The survey parameters are summarised in Table 1.

The vibroseis source generated linear sweeps of 24 s in duration and a 6–150 Hz frequency range. The DAS interrogator (Silixa iDAS v2, Hertfordshire, UK) was set to a 10 m gauge length and a 5 m pulse length with a spatial sampling interval of 1 m and temporal sampling of 1 ms. In such settings, random noise related to the recording hardware should be the main receiver-side factor affecting the repeatability, while mispositioning is the main source-side repeatability factor.

Raw data records were correlated with a synthetic sweep. To assess repeatability, we made a pairwise comparison of all 76 shot points. For a given pair of shots, we calculated the NRMS value in a 60 ms running window for every pair of traces (Figure 6d). Then, we estimated the average repeatability along the direct P wave (Window A, ±30 ms around the first breaks, the light blue window in Figure 6) and a larger window, B, starting from the first breaks 400 ms down (the red window in Figure 6), which includes the direct P wave, P wave reflections, and multiples.

Two shot points located about 1 m apart (Figure 6) showed quite good repeatability for window A (20–25% NRMS), but poorer for the larger window B—about 40–50% NRMS. It appears that the non-repeatable noise for the larger window was mainly created by surface-related multiples, which can be more sensitive to mispositioning.

Comparing the central shot point with all others revealed uneven spatial distribution in window A repeatability (Figure 7a). The near-central shot points (<1–2 m) were quite repeatable (NMRS ~20%), while distant shot points were not (NRMS > 40%). However, the repeatability dropped to an NRMS of 120–140% when moving 7 m to the west from the centre, while it remained in the range of 40–60% NRMS when moving the source to the east. This zonal distribution of the direct-wave repeatability was likely controlled by significant lateral variations in the near-surface conditions. The same pattern was observed for window B (Figure 7d) but was less pronounced. Other shot point comparisons (Figure 7b,c) also revealed the zonal distribution of repeatability. However, the east-west zonal distribution was still noticeable. In this case, the average repeatability level was worse mainly because short-term multiples and reflected waves were more sensitive to misposition, as can be seen by comparing the repeatability of shot points 6 and 18 m apart (Figure 8).

A shot point from the western zone was compared to two shot points from the east zone located 6 m away and 18 m away from the west-like zone and had similar average repeatability. However, the closest shot points had an even distribution of repeatability along the well (Figure 8a). In contrast, the farthest shot point had poor repeatability at the top of the well, increasing with depth (Figure 8b). This is likely because shallow-depth wavefield travel times are affected more by strong mispositioning than deeper wavefield travel times. This effect is more pronounced for near-offset shot points.

A pairwise comparison of all 76 shot points (Figure 9) revealed trends in repeatability versus mispositioning in the range from the first cm to about 30 m. Average window A repeatability (Figure 9a) was expectedly better (~45% NRMS) compared to window B repeatability (~95% NRMS) (Figure 9b) for the misposition range from 0.5 to 20 m. However, the direct wave was slightly less immediately affected by mispositioning (the red trend in Figure 9) but had a substantial spread because of near-surface variations. The latter created a spread in NRMS distribution, which increased significantly with the increase of misposition, starting from 20–60% NRMS at 1 m and reaching 20–160% NRMS at 10 m (Figure 9a). For window B, we observed a more pronounced trend in repeatability decrease with misposition starting from about 60–100% NRMS at 2 m and reaching 110% NRMS at 20 m, with the spread of ±30% NRMS likely attributable to spatial near-surface variations.

## 4. Discussion

While the primary focus of our study was the effect of mispositioning, repeatability can be affected by other factors. As shown in the Otway experiment, one such cause is temporal variations of near-surface conditions. Spatial near-surface variations can also strongly degrade repeatability, even for small source positioning variations. More generally, the way the survey is conducted can affect the data. For instance, the shape of the vibroseis plate has a particular directivity pattern, and thus changes in its azimuthal orientation, or up-hill/down-hill position, can degrade repeatability.

Our experiments are limited to only two geographical locations. The characteristic of spatial variations (homogeneous/heterogeneous, seasonal changes, among others) likely vary from site to site and depend on the local soil properties. The Otway site has sandy-clayey soil, while the Curtin site has a predominantly sandy surface. Systematic series of experiments on different types of surfaces are needed to study the effect on repeatability in more detail.

Our study was limited to vibroseis sources widely used in seismic on-shore exploration. The same factors, such as lateral variations of the near-surface, should affect the repeatability of other land sources, such as weight drop or explosives. However, there also may be source-specific effects. Lateral variations of soil conditions (wet/dry) can affect how a weight drop bounces off the landing plate. These soil conditions can also affect the directivity and strength of an explosion. These effects are source-specific and require additional studies.

Our study involved same-day experiment and different-day different-season experiments. However, it is important to understand what would happen if an experiment was conducted in different years in the same season. Jervis, Bakulin, Burnstad, Berron and Forgues [10] demonstrated same-day variations of repeatability reaching 1 ms and a 20% amplitude change. Half a year of permanent source recordings showed seasonal repeatability variation, highlighting rapid repeatability decrease on rainy days [23]. Thus, even small-time intervals such as days or hours can lead to temporal variations of repeatability.

## 5. Conclusions

Two experiments were conducted to assess the effect of source mispositioning on 4D DAS VSP repeatability. The experiment at the Otway site demonstrates that the effect of source mispositioning on repeatability is negligible in comparison with the effect of temporal variations of the near-surface conditions. To avoid these limitations, we conducted a same-day controlled experiment at the Curtin University site. This second experiment showed that the effect of source mispositioning on repeatability was controlled by the degree of lateral variations of the near-surface conditions. Unlike in marine seismic measurements, spatial variations of near-surface properties can be laterally rapid and would degrade the repeatability for even a few meter shifts of the source. The greater the mispositioning, the higher the chance of such significant variations. When the near-surface conditions are laterally homogeneous, the effect of modest source mispositioning is small, and in all practical monitoring applications, its contribution to non-repeatability is negligible.

## Figures and Tables

**Figure 1 sensors-22-09742-f001:**
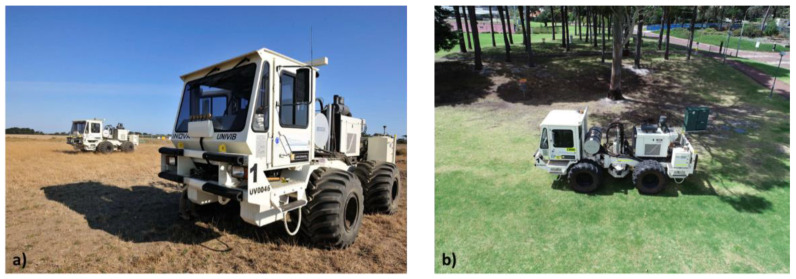
Photos of vibroseis truck at Otway (**a**), and close to NGL facility (**b**). Notice different soil conditions at the two sites. Sandy-clay soil at Otway is hard when dry and soft when wet, while sandy soil at NGL is less susceptible to moisture.

**Figure 2 sensors-22-09742-f002:**
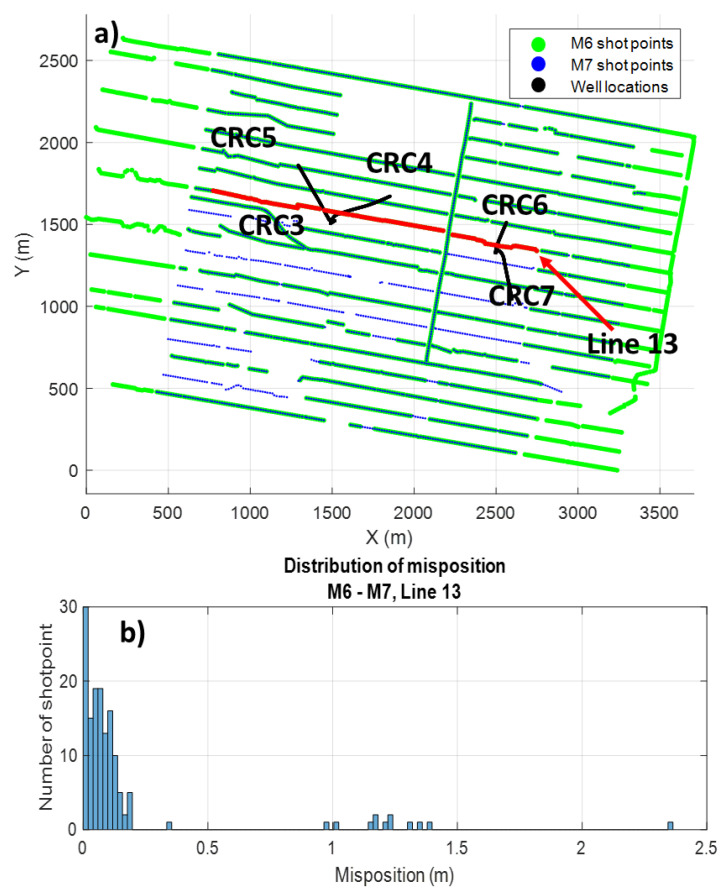
M6 and M7 acquisition map (**a**), and distribution of misposition of Line 13 (**b**). Green and blue dots indicate M6 and M7 survey shot points, respectively, and black lines are the horizontal location of CRC3–CRC7 monitoring wells. Red dots represent shifted Line 13.

**Figure 3 sensors-22-09742-f003:**
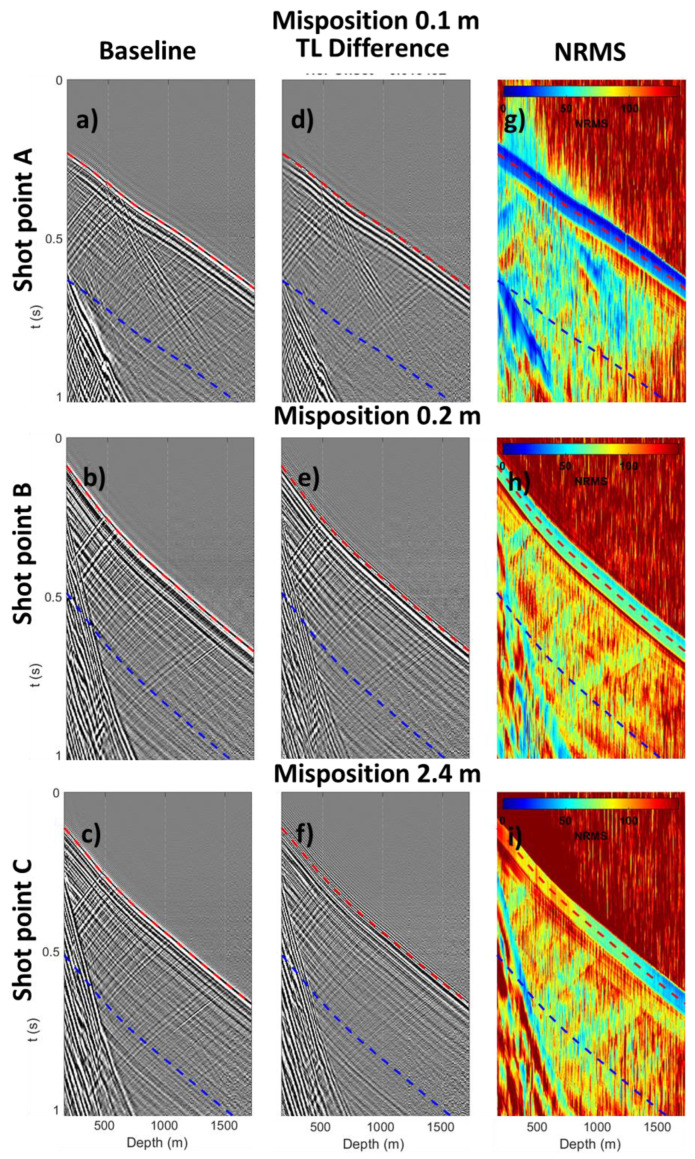
Comparison of shot points with relatively low and high misposition. Baseline shot points (**a**–**c**), time-lapse differences (**d**–**f**), and NRMS maps (**g**–**i**) are shown for shot points A, B and C with 0.1, 0.2 and 2.4 mispositioning, respectively. Time-lapse differences images have ×5 image gain for visual purpose.

**Figure 4 sensors-22-09742-f004:**
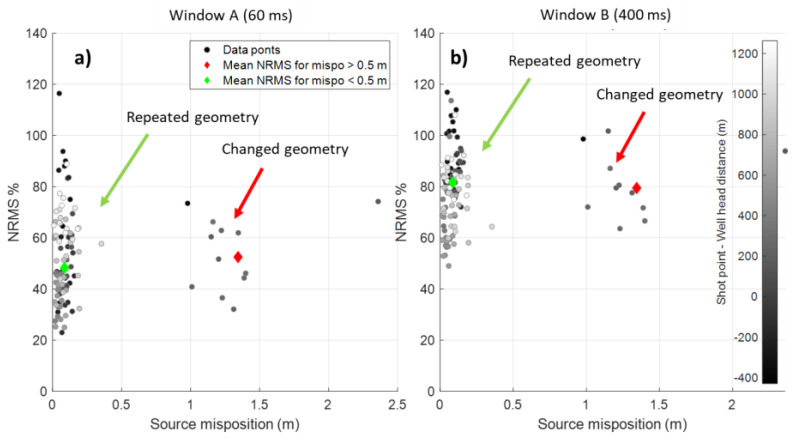
Distribution of repeatability for line 13 computed for a direct wave 60 ms window (**a**) and a P wavefield 400 ms window (**b**). Points are colour-coded by distance from the wellhead. A negative offset means the shot point is located east of the wellhead. Black and red diamonds are average misposition for ‘normal’ and mispositioned shot point groups.

**Figure 5 sensors-22-09742-f005:**
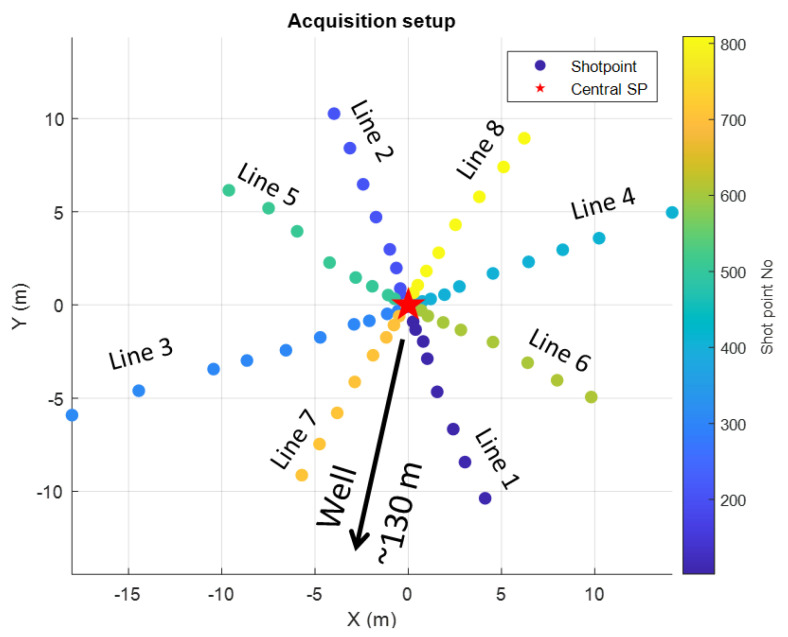
The acquisition map. Each line starts at the central shot point (red star). The well is 130 m to the south from the central shot point.

**Figure 6 sensors-22-09742-f006:**
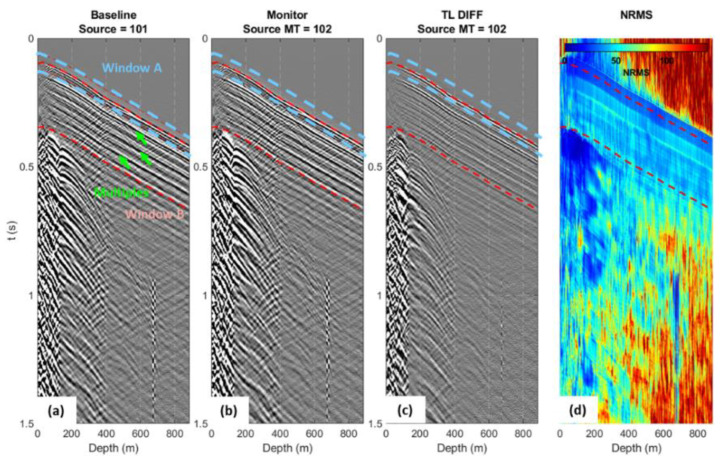
Two shot points located 0.93 m apart (**a**,**b**), their difference (**c**), and NRMS map (**d**). Computation windows for average NRMS are shown on (**a**): light blue window A includes direct P wave, and red window B includes direct P wave, reflected and multiple waves. Green arrows point to multiple waves.

**Figure 7 sensors-22-09742-f007:**
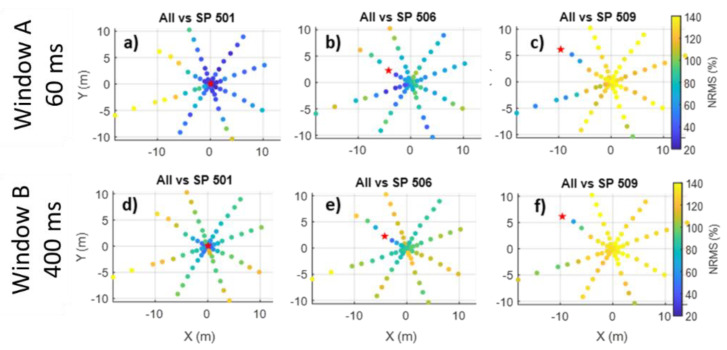
Comparison of one versus all shot points repeatability for central, middle, and far shot points of Line 5 for window A ((**a**–**c**) respectively, 60 ms window) and window B ((**d**–**f**) respectively 400 ms window). Shot points are colour-coded with NRMS value when comparing a red-star shot point with a given one. Notice a significant difference in repeatability between eastern and western regions.

**Figure 8 sensors-22-09742-f008:**
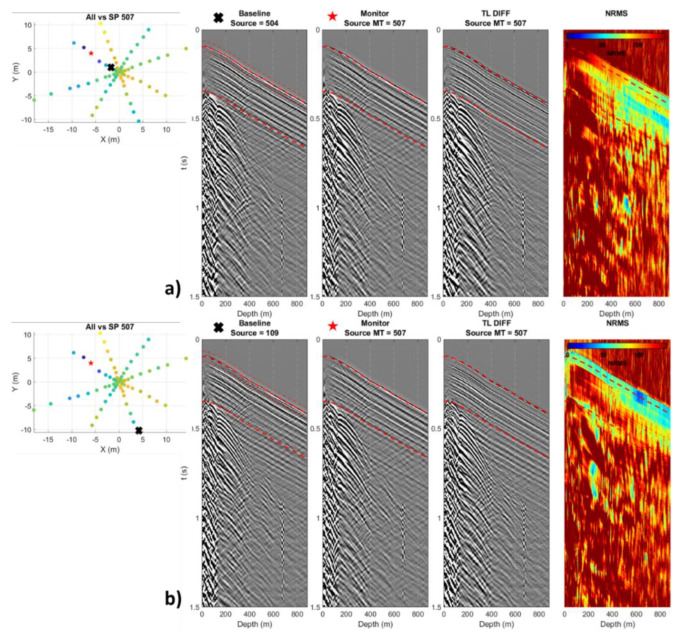
Selection of two shot points located 6 m (**a**) and 18 m (**b**) away. The shot points have similar repeatability levels and a significant difference in misposition.

**Figure 9 sensors-22-09742-f009:**
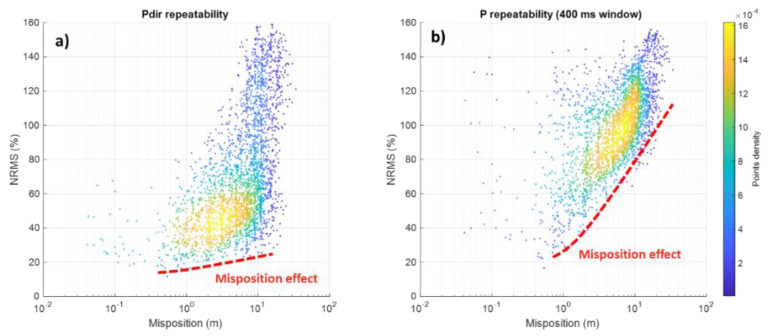
Distribution of repeatability (NRMS) vs. misposition for window A (**a**) and window B (**b**) for pairwise comparison of 76 shot points. Points are colour-coded by density distribution. The red dashed line highlights a possible misposition effect on repeatability. The spread is likely to be attributed to the spatial near-surface variations.

**Table 1 sensors-22-09742-t001:** Survey parameters.

Parameter	Otway M6	Otway M7	NGL
Survey date	March 2020	January 2021	May 2021
Source	Vibroseis INOVA UniVib 26,000 lbs	Vibroseis INOVA UniVib 26,000 lbs	Vibroseis INOVA UniVib 26,000 lbs
Sweep	Linear 6–150 Hz	Linear 6–150 Hz	Linear 6–150 Hz
Number of Source Positions	4084	3085	76
Shot spacing (m)	15	15	0.5–5
Fiber optic cable installation	Cemented behind the casing	Cemented behind the casing	Cemented behind the casing
Type of fibre	Constellation	Constellation	Single mode
DAS interrogator	Silixa iDAS v3	Silixa iDAS v3	Silixa iDAS v2
Spacing between virtual receivers (m)	5	5	1
Well depth (m)	1600	1600	900

## Data Availability

Not applicable.

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
