# Peer review of "Effect of Source Mispositioning on the Repeatability of 4D Vertical Seismic Profiling Acquired with Distributed Acoustic Sensors"

_sensors, 2022, doi:10.3390/s22249742_

Round 1

Reviewer 1 Report

This work of source mispositioning on the repeatability of 4D vertical seismic profiling is still very important to the field of earth geophysical exploration. I think this will promote the future application and popularization of DAS technology in this field, especially in terms of detection accuracy and reserve estimation. Therefore, this is a very good research work. I have a few small suggestions for the author:

1. DAS technology is a 1-dimensional system detection technology. How does it achieve 4D vertical seismic profiling positioning? What is its own positioning accuracy? Is it itself positioned far greater than the source mispositioning?

2. This article mainly focuses on two experiments to verify the effect of source mispositioning on the repeatability of 4D vertical seismic profiling, but how the DAS technology implementation is not described in the text, and needs to supplement the relevant principles.

3. What are the other shortcomings in this technology? How to compare it with the existing technology?

Author Response

Thank you very much for the comments. Please see the reply in the attachment.

Reviewer 2 Report

All the figures are verry clear. Therefore, the only minor revision is necessary.

1)     Figure 1a) : The ground condition is not clear. Please indicate the position of reservoir.

2)     Hopefully, a photo of observation is helpful for the reader to understand the research site.

Author Response

(The authors gave the same response as above.)

Reviewer 3 Report

Based on the vertical seismic profile and distributed acoustic sensing, the author of this paper explores the influence of wrong location, season and other factors on the detection repeatability. Combining the actual experiment with the theory, the research and verification have been carried out, and the expected conclusion has been obtained, which is innovative, but there are still the following problems to be revised.

1. Although there are two experimental sites, they seem to be independent experiments. Can we use different experimental sites to increase the validation of other factors that affect repeatability, such as geological conditions?

2. Influencing factors in the experiment are numerous and complex. Could we add qualitative charts/tables for various factors to facilitate better reading and increase the rigor of the experiment?

3. The experiment involved too few influencing factors and had certain limitations. Whether to consider adding other experiments to make the conclusions more extensive and practical

Author Response

(The authors gave the same response as above.)
